# Hyper-Crosslinked Porous Organic Nanomaterials: Structure-Oriented Design and Catalytic Applications

**DOI:** 10.3390/nano13182514

**Published:** 2023-09-08

**Authors:** Yiqian Luo, Yixuan Mei, Yang Xu, Kun Huang

**Affiliations:** 1School of Chemistry and Molecular Engineering, East China Normal University, Shanghai 200241, China; 51214300130@stu.ecnu.edu.cn; 2MOE Key Laboratory of Macromolecular Synthesis and Functionalization, Department of Polymer Science and Engineering, Zhejiang University, Hangzhou 310058, China; meiyixuan@zju.edu.cn; 3School of Materials Science and Engineering, Tongji University, Shanghai 201804, China

**Keywords:** hyper-crosslinked polymer, porous organic materials, catalyst, morphology, pore size

## Abstract

Hyper-crosslinked porous organic nanomaterials, especially the hyper-crosslinked polymers (HCPs), are a unique class of materials that combine the benefits of high surface area, porous structure, and good chemical and thermal stability all rolled into one. A wide range of synthetic methods offer an enormous variety of HCPs with different pore structures and morphologies, which has allowed HCPs to be developed for gas adsorption and separations, chemical adsorption and encapsulation, and heterogeneous catalysis. Here, we present a systematic review of recent approaches to pore size modulation and morphological tailoring of HCPs and their applications to catalysis. We mainly compare the effects of pore size modulation and morphological tailoring on catalytic applications, aiming to pave the way for researchers to develop HCPs with an optimal performance for modern applications.

## 1. Introduction

Porous organic nanomaterials, also known as hydrocarbons, are characterized by the presence of pores or voids. According to the IUPAC recommendations, pores in these materials, that have continuous pathways connecting them to the outer surfaces of the porous structure, are referred to as open pores [1]. In contrast, pores that are isolated or not connected to other pores are referred to as closed pores [2]. Porous organic nanomaterials have good stability, chemical resistance, and excellent hydrothermal resistance, which can be effectively applied in gas separation, catalysis, energy storage, and other important applications [2].

Currently, porous organic nanomaterials can be classified into covalent organic frameworks (COFs) [3,4,5], hyper-crosslinked polymers (HCPs) [6,7,8,9], conjugated microporous polymers (CMPs) [10,11], polymers of intrinsic microporosity (PIMs) [12,13], covalent triazine frameworks (CTFs) [14,15], and porous aromatic frameworks (PAFs) [16,17,18], depending on their composition and bonding nature. Among them, hyper-crosslinked polymers (HCPs) are a family of permanently microporous polymeric materials originally discovered by Davankov and which have gained increasing interest [19]. HCP is characterized by simple synthesis methods, low cost, a wide choice of monomers, high yields, and easy post-modification [20,21]. Due to their relatively stable porous structure and resistance to collapse, HCPs have been widely used in catalysis [22,23], adsorption [24], sensing [25,26], drug delivery [27,28], gas storage [29], and other fields [30,31,32]. To date, there are already reports on the comprehensive review of the applications of hyper-crosslinked polymers (HCPs) in areas such as biomedical applications [33], removal of heavy metal ions [34], and dye adsorption [35]. Especially, the HCPs with multifunctional groups and good chemical stability are a good choice as platforms for heterogeneous catalysis [36,37].

In recent years, massive studies have focused on pore size modulation and morphological tailoring of HCPs in order to further optimize and enhance their performance for catalytic applications (Figure 1) [6,38,39]. For example, the modulation of multi-stage pores can improve the efficiency of mass transport and enhance catalytic activity; the tailoring of the macroscopic morphology can enable selective and complex catalytic applications, etc. Previous studies have explored the impact of hierarchical pore structures on catalytic performance [40]. Additionally, the literature has documented the synthesis methods for creating shape-controllable hyper-crosslinked polymer nanoparticles [41]. However, existing reviews typically focus only on one aspect, either the catalytic applications or the synthesis methods, rarely delving into the deep connections between the two, especially the effects of pore structures and morphologies on catalytic performances [21,42,43]. The summaries of the specific mechanisms by which pore size modulation and morphological tailoring of HCPs affect catalytic performance are still scarce and require further discussion.

Here, we present a systematic review of recent approaches to pore size modulation and morphological tailoring of HCPs and their applications to catalysis, with a focus on the effect of pore size modulation and morphological tailoring on catalytic applications. The review is divided into the following sections: (1) synthesis methods for hyper-crosslinked polymer nanomaterials; (2) structure-oriented design of hyper-crosslinked polymer nanomaterials; (3) the effects of pore size modulation and shape tailoring of HCPs on catalysis; (4) conclusions and perspective.

## 2. Synthesis Methods for Hyper-Crosslinked Polymer Nanomaterials

Hyper-crosslinked polymers are typically synthesized using different techniques, including post-crosslinking [42], direct crosslinking [43], and crosslinking with the addition of crosslinkers (Figure 2) [44]. These diverse methods enable the formation of highly crosslinked polymer networks with unique structural and functional properties, allowing for a wide range of applications in fields such as materials science, catalysis, and separation technologies.

### 2.1. From Small Molecules

The synthesis of hyper-crosslinked polymers from small molecules is usually accomplished by self-crosslinking strategies or by the addition of crosslinkers.

#### 2.1.1. Self-Crosslinking Strategy without Additional Crosslinker

In general, self-crosslinking strategies rely on the inherent capability of monomers to polymerize and form crosslinked structures directly. For instance, the direct polymerization of multifunctional monomers like phenolic resins and dense amine resins has been extensively researched [46,47,48]. Furthermore, self-crosslinking methods employing oxidative coupling reactions and Friedel–Crafts alkylation have also been documented in the literature [44,49].

Han and his team have successfully synthesized a series of hyper-crosslinked porous poly(carbazole) materials, designated as CPOP-22~27, through a one-step synthesis using carbazole monomers (Cz22~27) containing carbonyl groups, employing oxidative coupling polymerization and Friedel–Crafts reaction (Figure 3a) [50]. The Brunauer–Emmett–Teller (BET) specific surface areas of CPOP-22~27 range from 440 to 760 m^2^·g^−1^.

In 2022, Yang et al. also utilized the Friedel–Crafts alkylation reaction of an ionic liquid monomer and 4,4′-bis(chloromethyl)-1,1′-biphenyl (BP) to prepare a series of PIPs (Figure 3b) [51]. This catalyst enables efficient CO_2_ cycloaddition reactions under metal-free, co-catalyst-free, and solvent-free conditions, exhibiting both universality and cycling stability.

#### 2.1.2. External Crosslinker Strategy

During the synthesis process, the direct addition of crosslinking agents such as formaldehyde dimethyl acetal (FDA) [52], sulfur, and divinylbenzene can also facilitate the one-pot synthesis of hyper-crosslinked polymers. It has been demonstrated that the direct interweaving of nucleophilic aromatic monomers in the presence of electrophilic crosslinking agents, such as formaldehyde dimethyl acetal, is an economically efficient and widely applicable method for synthesizing porous organic polymers.

In 2023, Lu et al. achieved the one-pot synthesis of halogenated ionic polymers (HIPs) by employing histidine, α,α′-dibromo-p-xylene (DBX), or α,α′-dichloro-p-xylene (DCX) as crosslinking agent under Lewis acid catalysis (Figure 4a) [53]. HIPs are hyper-crosslinked ionic polymers that incorporate nucleophilic halide ions into the HCP framework. During the reaction process, quaternization and Friedel–Crafts alkylation reactions occur simultaneously, resulting in a polymer framework with abundant hierarchical porosity and ionic sites. The synergistic effects of carboxyl groups, amino groups, halide ions, and secondary amines within the polymer framework enable efficient CO_2_ cycloaddition reactions under metal-free, co-catalyst-free, and solvent-free conditions. These HIPs exhibit universality and cycling stability, making them promising candidates for various applications.

In 2021, Huang et al. designed three novel poly(ionic liquid) materials composed of one, two, and four benzene rings, utilizing 2-phenylimidazole as the building block (Figure 4b) [54]. These poly(ionic liquid) materials were synthesized by polymerizing α,α′-dichloro-p-xylene (DCX) and formaldehyde dimethyl acetal (FDA) with the designed ionic liquid (IL) monomers. The self-crosslinking of the crosslinking agent contributed to the large specific surface area of the hyper-crosslinked organic frameworks, while the co-crosslinking between the crosslinking agent and the ionic liquid introduced active sites into the frameworks. The resulting porous hyper-crosslinked polymers exhibited a surface area of 763 m^2^·g^−1^ and demonstrated efficient selective absorption of CO_2_, along with remarkable activity in the cycloaddition reactions of CO_2_ and epoxides. These findings highlight the potential of these materials for applications involving high-performance CO_2_ capture and utilization.
Figure 4(**a**) Procedure for synthesis of HIP-X-His (X = Br or Cl). Reprinted with permission from Ref. [53], Copyright 2023, Elsevier. (**b**) HP-[BZPhIm]Cl–Co-Tols Formed by [BZPhIm]Cl and FDA, and HP-[BZPhIm]Cl-DCXs Formed by [BZPhIm]Cl and DCX. Reprinted with permission from Ref. [54], Copyright 2021, American Chemical Society.
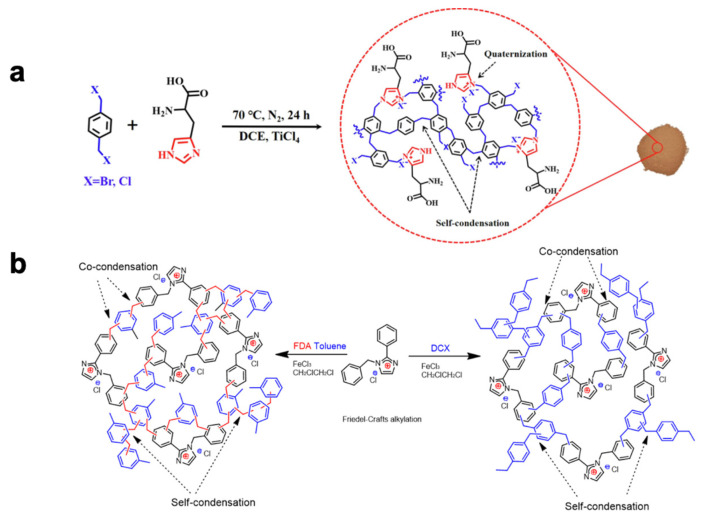



In 2023, Cao and Peng et al. reported a study involving the synthesis of hyper-crosslinked polymers (HCLP) using polycyclic aromatic hydrocarbons (PAHs) such as naphthalene, anthracene, pyrene, and quinoline as monomers [55]. The researchers employed an external crosslinker knitting method (ECLKM) and utilized fatty amines as nitrogen sources for impregnation to prepare the HCLP. The resulting HCLP exhibited exceptionally large specific surface areas (up to 2870 m^2^/g), narrow pore distributions (<0.70 nm), and high pore volumes (up to 1.09 cm^3^/g), owing to appropriate porosity and nitrogen doping.

### 2.2. From Macromolecules

Similarly, macromolecules can be used as precursors to achieve hyper-crosslinked polymers in the form of networked polymers by adding intermolecular covalent bonds. The general synthesis methods are post-crosslinking and crosslinking with the addition of crosslinkers [56,57].

In 2006, Tetley et al. reported the synthesis of gelatinous spherical polystyrene precursors with a diameter typically around 400 nm through emulsion polymerization without the use of surfactants [58]. Subsequently, hyper-crosslinking was performed between the vinyl groups to prepare polymer microspheres. This process resulted in the formation of highly porous organic polymers with a specific surface area of up to 1350 m^2^/g. Moran-Mirabal and his team have successfully addressed the challenge of impaired interparticle interactions and network formation in cellulose nanocrystals (CNCs), which are hydrolysis products of cellulose [59]. By modifying the CNCs, they have achieved hyper-crosslinked nanomaterials, overcoming the issue of ineffective CNC interaction and network structure formation caused by the introduction of sulfate hemi-ester groups onto the CNC surface during the hydrolysis process using sulfuric acid. They have proposed a method for producing stable spherical cellulose particles through the chemical crosslinking of aldehydes and hydrazide-modified CNCs. This method offers a solution to generate chemically stable cellulose microspheres with enhanced interparticle interactions and network structures.

In 2020, Wu’s team reported a study on the fabrication of yolk-shell nanospheres with silver nanoparticles encapsulated within porous polymer hollow spheres [60]. They synthesized PS-b-PAA block copolymers and obtained highly crosslinked polymer nanospheres through Friedel–Crafts alkylation (Figure 5a).

Huang and his team utilized polylactide-b-polystyrene (PLA-b-PS) diblock copolymer as a precursor to fabricating hollow microporous organic nanospheres (H-MONs) through a hyper-crosslinking-mediated self-assembly strategy [61]. In this approach, the hyper-crosslinked PS segment formed a microporous organic shell framework, while the degradable PLA segment generated a hollow mesoporous core structure. In 2019, they also prepared a metal nanoparticle-encapsulated hollow porous polymeric nanosphere framework M@HPPNF (M = Pd, Ru, or Pt) catalyst with a surface area of 630 m^2^/g using a polylactide-b-polystyrene (PLA-b-PS) diblock copolymer precursor (Figure 5b) [62]. The catalyst exhibited good activity and tunable size selectivity in the hydrogenation reaction.
Figure 5(**a**) Schematic illustration for fabrication of Ag@PAA-xPS and its applications to catalysis. Reprinted with permission from Ref. [60], Copyright 2020, Chinese Chemical Society Institute of Chemistry, Chinese Academy of Sciences Springer-Verlag GmbH Germany, part of Springer Nature. (**b**) Proposed synthesis process of M@HPPNFs (M = Pd, Ru, or Pt), TEM micrograph of Pd@HPPNFs. Reprinted with permission from Ref. [62], Copyright 2019, American Chemical Society.
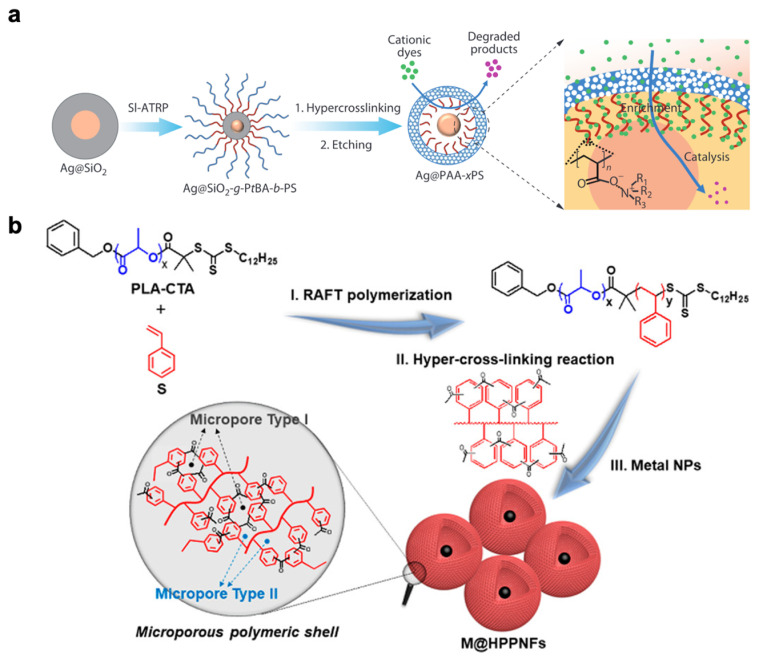



## 3. Structure-Oriented Design of Hyper-Crosslinked Polymer Nanomaterials

Due to the monomeric nature of hyper-crosslinked polymers and the variety of methods used to synthesize them, an increasing number of studies focus on designing their pore structures and morphology to achieve better performance. Therefore, it is important to study and summarize the design rules in order to obtain the optimum structure more systematically and easily in subsequent studies.

### 3.1. Design Methods for Pore Size Modulation and Distribution

It is well known that pore sizes < 2 nm are known as micropores, while those between 2 nm and 50 nm are known as mesopores. The degree of crosslinking of a hyper-crosslinked polymer largely affects its pore size and volume. Pore size design can also generally be achieved by artificially adding pore-forming agents, templates, etc. The most common methods are the hard and soft template methods. Hard template methods such as, for example, Ag [60] or silica [63,64], can be followed by etching of the template with strong acids and other reagents to form a porous structure. The soft template method often adds surfactants as porogenic agents in the polymerization process, and the removal of porogenic agents is generally greener, they can be removed by solvent washing and other means [65,66]. Both of these methods form mostly macro/mesopores.

The hyper-crosslinking of suitable monomers can result in the formation of a microporous structure. When we choose monomers with a rigid structure, such as benzene rings, imidazole, and carbazole, which contain five-membered or six-membered rings, these rigid structures can form microporous structures through inter-crosslinking or crosslinking with crosslinking agents [50,67]. Additionally, the rigid structures also impact the stacking state of polymer chain ends and, by modifying their stacking state, the pore size structure can be adjusted [68]. Interestingly, these approaches not only modify the pore size and distribution of the hyper-crosslinked polymers but also influence their catalytic performance.

The method of removing functional groups from the polymer framework through etching and breaking of chemical bonds can adjust the size and number of micropores in the hyper-crosslinked polymer. In 2017, Li and colleagues reported a study on the preparation of novel porous materials using PMS as a starting material [69]. They crosslinked the benzene moieties on PMS through FDA to construct a HCP (highly crosslinked polymer) with moderate surface area. Then, they etched the Si-O bonds using hydrofluoric acid (HFA) to create additional pores within the HCP. Simultaneously, some uncrosslinked phenyl groups were removed, adjusting the conformation of the remaining polymer skeleton to generate micropores and mesopores. For samples with a feed ratio of 1.5, the number of micropores and mesopores in F-HCP-1.5 within the range of 1.0 to 9.3 nm was significantly higher than the number in HCP within the corresponding range (Figure 6a). The extra micropores and mesopores were generated by removing PMS segments. Compared to the micropore and mesopore volumes of HCP, F-HCP exhibited an increase in micropore and mesopore volumes. Therefore, by adjusting the crosslinking and etching conditions, it is possible to easily modulate surface area, pore size distribution, and other properties.

In 2023, Wang et al. reported a study on the synthesis of chiral porous hyper-crosslinked polymer frameworks using L-phenylalanine dipeptide as a monomer through a simple Friedel–Crafts alkylation reaction. The resulting framework was named HCP(L-Phe-L-Phe-OMe) [70]. Subsequently, they further modified the framework using L-proline. The BET surface area of the L-Pro-modified sample experienced a notable reduction, declining from 661.74 m^2^/g (HCP(L-Phe-L-Phe-OMe)) to 261.68 m^2^/g (HCP(L-Pro-L-Phe-L-Phe-OMe)). Additionally, there was a decrease in pore volume, which dropped from 0.52 cm^3^/g to 0.15 cm^3^/g, while the pore size exhibited a slight increase, growing from 6.23 nm to 6.76 nm (Figure 6b). The reaction time of cyclohexanone and trans-β-nitrostyrene in the Michael reaction was reduced from 7 days to 4 days, and the enantiomeric excess (ee) value increased from 71% to 97%. Short chiral peptides in HCP mesoporous frameworks are believed to provide unique and stable chiral environments within the pores. The presence of the -Pro residue within the pores catalyzes and restricts the reaction, resulting in the catalyst exhibiting high enantioselectivity. In 2022, Patra and their team selected tricyclic monomers to create highly porous networks due to the paddlewheel-like topology and “internal free volume” of tricyclic monomers, which result in inefficient stacking of polymer chains [68]. They synthesized irregular polymers (FCTP), rigid spheres (SCTP), and two-dimensional nanosheets (SKTP) using three different methods. Among them, SCTP exhibited a pore size distribution at the ultra-micropore domain (0.64 nm), suggesting that the pores at 0.64 nm might be attributed to the inefficient stacking of two tricyclic cores (Figure 6c). Additionally, the presence of spiral aromatic-type pore structures may lead to pore sizes of approximately 0.9 nm.

Modifying functional groups can also achieve the purpose of altering the distribution of mesopores. In 2022, Tan et al. reported on the successful introduction of hydrazine (HZ) through post-modification into metalloporphyrin-based hyper-crosslinked polymers, resulting in the generation of abundant CO_2_ chemical adsorption sites within the structure [71]. This enabled efficient adsorption (with a Qst value of up to 34.7 kJ/mol) and chemical transformation of carbon dioxide. HCP-TPP, HCP-TPP-SO_3_H, and HCP-TPP-Co-SO_3_H exhibited a microporous structure with primary pore sizes below 2 nm (peaks at 0.8 nm, 1.1 nm, and 1.5 nm), with some mesopores at 2.1 nm, indicating a hierarchical pore size distribution. After grafting with amino groups, the mesopores were blocked, and the main pore sizes of HCP-TPP-Co-HZ were concentrated around 0.5 nm and 1.3 nm. HZ-modified material (HCP-TPP-Co-HZ) exhibits high catalytic performance towards epoxy substrates of different molecular sizes, with excellent yields (reaching 98% for epichlorohydrin). This is significantly higher than HCP-TPP-Co-SO_3_H (64%), which can be attributed to the chemisorption sites enriched with CO_2_ reactants.

In 2016, Dai and his team designed and synthesized cyclodextrin-based hyper-crosslinked polymers, which were then applied to the selective adsorption and storage of CO_2_ [72]. All of these materials exhibited both microporous and mesoporous features in CD-based HCPP, likely resulting from the irregular stacking of rigid and twisted CD (cyclodextrin) molecules as well as the inherent cavities within the CD structures. HCPPs based on αCD (alpha-cyclodextrin) displayed more concentrated micropores compared to their counterparts based on βCD (beta-cyclodextrin). Noticeable reductions in pore size, particularly around the 1 nm range, were observed, demonstrating the feasibility of manipulating cyclodextrin cavity sizes to control micropore distribution.

In 2018, Sun and colleagues utilized glycidyl methacrylate-shrinkable glycerol ester-polyethyleneimine monomers (GMA-PEI) and different crosslinking agents to obtain a series of microporous organic polymers linked with polyethyleneimine (PEI) [73]. The gas adsorption capacity of the NUT polymers followed the order of NUT-10 > NUT-9 > NUT-8, primarily due to differences in amine content and structural properties. NUT-10 exhibited a higher amine content and a larger microporous surface area, resulting in better CO_2_ adsorption capabilities.

Mesopores are more easily formed by introducing a template or changing the degree of crosslinking between hyper-crosslinked polymers. In 2015, Wang et al. reported a study in which Fe_3_O_4_ superparticles were used as the core, and micro/mesoporous polyoctylene pimelate (POP) served as the shell [74]. They synthesized Fe_3_O_4_@PS microspheres with a core/shell structure and initiated solvation polymerization in the PS shell using a mixture of divinylbenzene (DVB) and vinyl chloride (VBC). The resulting poly(VBC-*co*-DVB) network mixed with PS in the shell, leading to significant phase separation. After undergoing a Friedel–Crafts type hyper-crosslinking treatment, POP structures with specific porosity were obtained. During this process, non-crosslinked PS chains were extracted from the shell, and large mesopores were achieved through the microporosity obtained by the hyper-crosslinked network. By varying the feed volumes of VBC and DVB, the phase separation was altered, allowing for the adjustment of major mesopores from 4 nm to 30 nm.

In most cases, mesopores and micropores are simultaneously adjusted during the synthesis of hyper-crosslinked polymer nanomaterials. In 2018, Červený et al. reported the synthesis of PPC-type conjugated hyper-crosslinked poly(arylethynylene) networks through chain-growth copolymerization using 1,4-diethynylbenzene, 1,3,5-triethynylbenzene, and tetra(4-ethynylphenyl) methane. These PPC materials exhibit permanent microporous/mesoporous structures and a high specific surface area (S_BET_) of up to 1000 m^2^/g. The PPC materials demonstrate activity in acid-catalyzed reactions such as aldehyde and ketone condensation and carboxylic acid esterification [75]. Rao et al. utilized benzene and triphenylphosphine as precursors and employed FeCl_3_ as a catalyst to achieve the Friedel–Crafts reaction, successfully synthesizing a porous phase change material (PCM) [39]. The material demonstrated the presence of numerous micropores and mesopores, exhibiting a maximum thermal conductivity of up to 600% and a high photothermal conversion efficiency of 93.7%. Firstly, the presence of silicon groups on the polystyrene backbone can lead to hyper-crosslinking reactions. Due to the cascade substitution and elimination reactions between silicon–oxygen groups, an increased proportion of silicon–oxygen groups will inevitably increase the degree of hyper-crosslinking in the original linear polymer, resulting in the formation of a porous structure with a larger specific surface area. In 2019, Li and his team developed a green and atom-economic strategy to construct microporous organic polymers (BMOPs) based on polybismaleimide [76]. In contrast to the conventional Friedel–Crafts alkylation reaction for HCP, they employed an alternative hyper-crosslinked polymerization technique based on free-radical copolymerization. The resulting product exhibited a high specific surface area and possessed a microporous/mesoporous structure.

In 2016, Huang and his team reported a novel synthetic method for amino-functionalized microporous organic nanotube networks (NH_2_-MONNs) [77]. This method involved hyper-crosslinking of the polystyrene shell in molecular bottlebrushes, followed by the removal of the retained PLA core within the pores. This process resulted in microporous organic nanotube networks with a trifold structure comprising micropores, mesopores, and macropores (Figure 6d).

In 2011, Wu et al. reported on the fabrication of nanostructured porous network (NPN) materials with three-dimensional (3D) interconnected nanochannels [78]. The strategy used well-defined hairy nanoparticles as building blocks for the network construction. Hairy nanoparticles, composed of polystyrene chains grafted onto the surface of silica nanoparticles via surface-initiated atom transfer radical polymerization (SI-ATRP), served as the core-shell units for network construction. Carbonyl crosslinking within and between the polystyrene particles was then employed. The final NPN materials contained three types of pores: (i) micropores induced by crosslinking of the hairy polystyrene shell, (ii) mesopores obtained by sacrificial removal of the silica nanoparticle cores, and (iii) mesoporous/macroporous network linkages formed through interparticle crosslinking (Figure 6e). Ouyang, Wu and colleagues have prepared a new powdered polymer aerogel (PPA) with a multistage pore structure by hyper-crosslinking monodisperse poly(styrene-co-divinylbenzene) nanoparticles [79].
Figure 6(**a**) Nitrogen sorption isotherms of (F-)HCP-1.5 and (F-)HCP-4.5 at 77.3 K, with pore size distribution shown from nitrogen sorption at 77.3 K using the NLDFT method. Reprinted with permission from Ref. [69], Copyright 2017, American Chemical Society. (**b**) (**A**) N_2_ adsorption–desorption isotherms of HCP(L-Phe-L-Phe-OMe) and HCP(L-Pro-L-Phe-L-Phe-OMe) and (**B**) pore size distribution curves of HCP(L-Phe-L-Phe-OMe) and HCP(L-Pro-L-Phe-L-Phe-OMe). Reprinted with permission from Ref. [70], Copyright 2023, Royal Society of Chemistry. (**c**) Pore size distribution of pristine HPOPs and sulfonated HPOPs based on the quenched solid density functional theory (QSDFT) method, with ultramicropore size distribution from CO_2_ sorption at 273 K/1.0 bar using the HK mode (inset). Reprinted with permission from Ref. [68], Copyright 2022, American Chemical Society. (**d**) N_2_ adsorption–desorption isotherms and pore size distributions calculated using the NLDFT method of NH_2_-MONNs (inset). Reprinted with permission from Ref. [77], Copyright 2016, Royal Society of Chemistry. (**e**) DFT pore size distributions of (**A**) SiO_2_/crosslinked PS composite, (**B**) nanoporous polystyrene, and (**C**) nanoporous carbon. Reprinted with permission from Ref. [78], Copyright 2011, American Chemical Society.
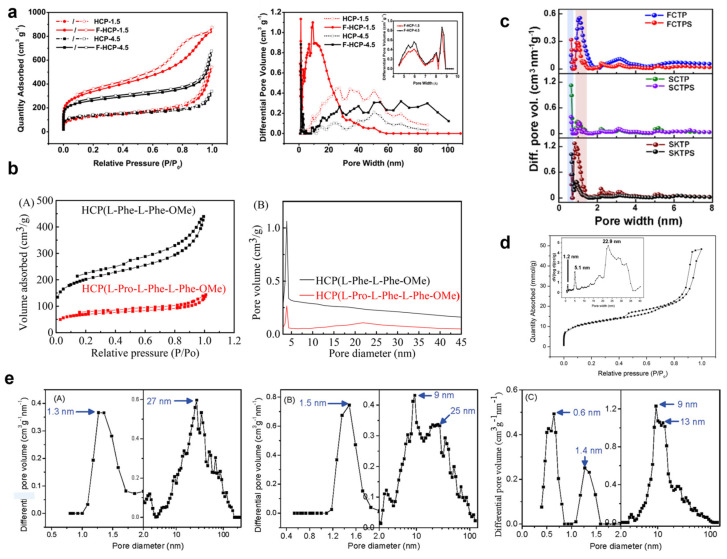



### 3.2. Design Strategies for Morphological Tailoring

Common catalysts with tailorized morphologies are generally yolk-shell, core-shell, honeycomb-like, and hollow tubular catalysts. By designing the morphologies of the HCPs, some special selective catalysis can be achieved by screening the reactants. The HCPs with hollow structure are more intriguing due to the presence of internal cavities. The internal cavities can serve as excellent enrichment sites, providing significant assistance in catalytic reactions.

In 2021, Tan et al. proposed a self-templating method for the preparation of monodisperse mesoporous hollow capsules (Figure 7a) [80]. By adding styrene and divinylbenzene (DVB) monomers at different stages of the polymerization reaction, dense core-shell spheres consisting of a pure polystyrene (PS) core and lightly crosslinked polystyrene-divinylbenzene (PS-DVB) shell were obtained in a one-pot synthesis. Control over particle size and hollow structure was easily achieved by varying the reaction conditions and the timing of DVB addition. These catalysts, immobilized with gold nanoparticles, exhibited excellent catalytic performance in the reduction of 4-nitrophenol model reaction. The exceptional catalytic performance is attributed to the catalyst’s high surface area and abundant hierarchical porosity. In 2022, they further advanced their work by synthesizing a new type of hyper-crosslinked polymer-based hollow organic microcapsules (PANI@S-HMOCs) [81]. This was achieved by in situ polymerizing aniline within the porous structure of sulfonated hollow mesoporous organic capsules (S-HMOCs), resulting in a modified shell composed of polyaniline (PANI).

Additionally, Tan’s team utilized divinylbenzene hyper-crosslinking to obtain microporous polymers and discussed the morphology, porosity, and applications of their derived hollow microporous carbon spheres (HCS) [6]. They controlled the various morphologies of core-shell microspheres by altering the content of divinylbenzene (DVB) and prepared HCS with eccentric core and central core configurations. It was demonstrated that the DVB content of the precursor hollow organic microporous carbon (HOMC) and the carbonization temperature played crucial roles in controlling the structural characteristics and maintaining the porous structure of HCS.

In 2022, Kim et al. prepared tunable-thickness sulfonic acid-modified hollow polymer nanospheres with a hollow structure generated through weak acid–base interactions originating from hydrogen bonding between monomers and catalysts [82]. Distinct weak acid–base interactions arising from Lewis acids and Brønsted acids influenced the dimensions, shell thickness, and surface topological structure. Catalysts with thinner shell thickness exhibited enhanced catalytic performance in the esterification reaction of lauric acid and coconut oil with MeOH, as the thinner shell thickness could promote diffusion during the reaction.

In 2019, the team of Sun and Zhao reported on the design of hyper-crosslinked polymer morphology of hollow mesoporous organic polymers (H-MOPs) [83]. They carried out post-synthetic modification on the hollow-type H-MOPs, resulting in a hyper-crosslinked polymer material called H-CMPL@HCP-BP. During the ε-caprolactone ring-opening polymerization to form polycaprolactone, H-CMPL@HCP-BP exhibited excellent heterogeneous catalytic performance and outstanding recyclability.

In 2018, Huang et al. reported a one-pot method for the synthesis of multifunctional hollow microporous organic nanospheres (H-MON) using a hyper-crosslinking-mediated self-assembly strategy. By introducing functional organic ligands, they successfully synthesized various metal-loaded or heteroatom-doped H-MONs. They demonstrated the advantages of the hollow structure by using the hydrogenation reaction of nitroaromatic compounds as an example, showing enhanced diffusion of reactants and products [84]. In 2019, Huang and Wang reported a strategy for synthesizing hollow porous polymer nanosphere frameworks (HPPNFs) as efficient yolk-shell catalysts. They found that solvents with different polarities have an influence on the swelling ratio of the nanospheres, and this swelling behavior is reversible. They demonstrated that the solvent polarity can be adjusted to control the free volume and thus regulate the micropore size of the HPPNFs shell, enabling size-selective catalytic reactions. Using the hydrogenation of cuparene modified with olefins in toluene or acetone as an example, they proved that the high swelling degree of the catalyst in toluene increased the micropore size and facilitated the transport of cuparene modified with olefins through the polymer shell of the nanoreactor [62].

In 2023, Shao et al. reported on the preparation of a magnetic hyper-crosslinked polymer composite material with a core-shell structure. They found that increasing the thickness of the shell affects monodispersity, leading to a decrease in specific surface area and hindering the accessibility of adsorption sites, thus resulting in slow mass transfer (Figure 7b) [85].
Figure 7(**a**) Synthetic route for fabrication of SMHCs using the self-templating strategy. TEM images of the hollow capsule precursors. Reprinted with permission from Ref. [80], Copyright 2021, Royal Society of Chemistry. (**b**) Preparation of Fe_3_O_4_@poly(MAAM-co-EGDMA) and its applications in the pretreatment of fat-rich foodstuffs (auto extraction system: 1 material activation, 2 lipids adsorption, 3 and 4 material elution). TEM image of Fe_3_O_4_ nanoparticles. TEM image of Fe_3_O_4_@poly(MAAM-co-EGDMA) with MAAM/EGDMA proportions at 5/1, 3/1, 1/1, 1/3, and 1/5. Reprinted with permission from Ref. [85], Copyright 2023, Elsevier.
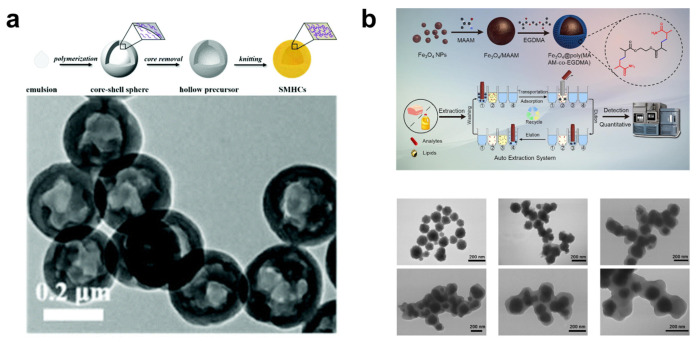



A honeycomb-like structure has also been found to be beneficial for mass transfer. In 2017, Huang’s team reported a strategy for the synthesis of honeycomb-like porous organic nanospheres using triphenylphosphine-guided hyper-crosslinking self-assembly. They further synthesized a catalyst encapsulating Pd nanoparticles through a simple impregnation-reduction method (Pd@HBP) (Figure 8a). The catalyst exhibited superior selective hydrogenation performance compared to similar homogeneous or heterogeneous counterparts, achieving a benzene amine separation yield of up to 99% in the hydrogenation of nitroaromatic compounds. They attributed this to the advantages of the three-dimensional (3D) honeycomb-like interconnected mesoporous structure, which allows accessible catalytic active sites to be efficiently exposed to the reactants, thereby promoting mass transfer more effectively. The size of the honeycomb-like mesopores could also be adjusted by varying the length of the degradable PLA domain [86].

Apart from the aforementioned structures, another interesting structure is hollow nanotubes. Huang and Rzayev firstly reported that the bottlebrush copolymers can act a as soft-template to form the cylindrical structures in solution [87]. By employing a crosslinking method, the original shape and size of the bottlebrush macromolecules can be preserved after core etching. In this approach, a hierarchical organic polymer tubular network can be directly obtained. Using this soft-template method, various functional groups such as amine, amino, sulfonic acid, porphyrin, and thiol can be incorporated or immobilized onto the tubular network with micropores, mesopores, and macropores, either in situ or post-synthesis [77,88]. They also synthesized a tubular nanomaterial with a hierarchical pore structure. By manipulating the distribution of micropores and mesopores within the structure, they confirmed the importance of the mesoporous structure for the highly efficient catalytic activity in the oxidation of benzyl alcohol. The absence of tubular mesopores might limit the mass transfer of reactants and products, as well as the accessibility of active sites [89].

In 2022, Liang et al. utilized hyper-crosslinked polymers to prepare bamboo-like carbon nanotubes with a hierarchical pore structure (Figure 8b) [90]. By comparing carbon nanotubes with different pore size distributions obtained through different preparation methods, they demonstrated the importance of hyper-crosslinked mesoporous channels for mass transfer and synergistic catalysis. The obtained hyper-crosslinked carbon nanotubes exhibited excellent catalytic performance in the photo-thermal-driven cycloaddition of CO_2_ with epoxides.
Figure 8(**a**) Synthetic route to Pd@HBPs from PLA-*b*-P(S/DPPS) diblock copolymer precursors (small balls represent Pd NPs). TEM images and SEM images of HBPs-1. Reprinted with permission from Ref. [86], Copyright 2017, American Chemical Society. (**b**) (**i**) Schematic illustration for the synthesis of Ni-BNCNTs@HMPs-NH_2_. (**ii**,**iii**) SEM images, (**iv**) TEM image, and (**v**) elemental mapping of Ni-BNCNTs. (**vi**) SEM image and (**vii**,**viii**) TEM images of Ni-BNCNTs@HMPs-NH_2_. (**ix**) Aberration-corrected HAADF-STEM image of the Ni-BNCNTs. (**x**) HAADF-STEM images and corresponding element mappings of Ni-BNCNTs@HMPs-NH_2_. Reprinted with permission from Ref. [90], Copyright 2022, Royal Society of Chemistry.
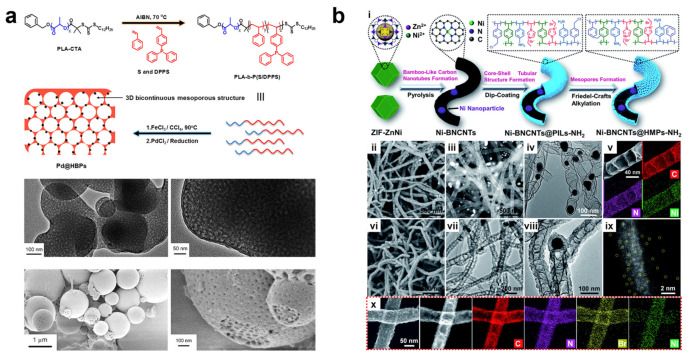



In this paper, we have explored the design strategies related to pore size and morphology of HCPs. It is evident from the preceding discussions that microstructural and macroscopic characteristics are not entirely independent of each other. By selecting suitable monomers or crosslinking conditions, the size and quantity of micropores can be regulated. The distribution of micropores can also be adjusted through subsequent modifications of the porous nanomaterial. On the other hand, the use of templating or external crosslinking agents allows for the control of mesopore size and distribution by varying the degree of crosslinking. Templating and self-assembly methods are the primary synthetic approaches for controlling the morphology. By selecting appropriate templates and polymerization methods, it is possible to simultaneously achieve the desired control over the pore size and morphology.

## 4. The Effects of Pore Size Modulation and Shape Tailoring of HCPs on Catalysis

The quantity and distribution of micropores and mesopores play crucial roles in shaping the overall macroscopic morphology of materials and their impact on catalytic performance. For instance, the internal cavities of mesoporous catalysts serve as mesoporous apertures that are interconnected with many micropores. The presence of micropores and mesopores greatly influences the mass transfer efficiency within the polymer structure, allowing for selective sieving of reactants and products by precisely tuning the pore size. The mass transfer efficiency directly affects the binding efficiency between the polymer and active sites, thus exerting a significant impact on catalytic performance. Furthermore, the existence of hollow structures in nanomaterials can serve as reservoirs, facilitating the enrichment of substrates and creating a more favorable environment for catalytic reactions. On a macroscopic level, the morphology of materials, such as yolk-shell, core-shell, honeycomb-like, and hollow tubular structures, plays a critical role in enhancing the contact between the material and reactants, thereby improving catalytic processes. By harnessing the unique morphologies of HCPs, such as hollow structures, we can achieve enhanced catalytic activity through shorter mass transfer pathways while preventing catalyst leakage. Simultaneously, the presence of porous structures within HCPs can prevent catalyst aggregation, thus avoiding a decline in catalytic activity.

In 2023, Lin et al. synthesized hyper-crosslinked polyionic liquids (HIPs) through Friedel–Crafts alkylation [91]. Their research revealed that the microporosity of this catalyst significantly influenced CO_2_ adsorption, and the ionic density had a certain impact as well. They evaluated the catalytic activity using a CO_2_ cycloaddition reaction, finding that HIPs with small specific surface areas or extremely low ionic content were not efficient in achieving high catalytic activity for CO_2_ cycloaddition reactions. Excellent catalytic performance required HIPs with high ionic content and large specific surface areas. This catalyst demonstrated outstanding performance in the cycloaddition reaction of CO_2_ and epoxides at 140 °C, 1 bar CO_2_ pressure, and 2 h reaction time. Under catalytic conditions, the product yield reached 99%, with a selectivity of 99%.

Peng and Lin, along with their colleagues, successfully synthesized porous imidazole-based hyper-crosslinked ion polymers with a large specific surface area (596 m^2^·g^−1^) and higher ion content (0.971 mmol/g) using 2-chloro-1,1-dimethoxyethane [92]. The synthesized hyper-crosslinked polymers exhibited a predominant pore size distribution in the range of 1–3 nm, indicating a hierarchical micro–mesoporous structure in all samples. The specific surface area (S_BET_) and pore volume (V_p_) of the hyper-crosslinked polymers were adjusted by varying the molar ratio of the monomer (BCB) and crosslinker (CDA). The use of the crosslinker CDA resulted in a significant increase in the Cl content of the hyper-crosslinked polymers, which was beneficial for subsequent quaternization reactions. The pore volume and specific surface area decreased with an increase in the ion content, as the increased ion content occupied more pore space. In the absence of co-catalysts, solvents, and additives, the catalyst exhibits high yield, selectivity, and good substrate compatibility in the cycloaddition reaction with CO_2_. The reaction occurs under atmospheric pressure with short reaction times (3–6 h) when reacting with epoxides. By appropriately extending the reaction time at low temperatures, a yield of 98% can be achieved under conditions of 100 °C and 20 h (Figure 9). This reaction time is shorter than that reported for most porous ionic catalysts under similar conditions in the past.

In 2020, Son et al. presented a novel post-synthetic strategy for microporous organic polymers (MOPs) based on AB_2_ polymerization chemistry (polymerization of AB_2_-type monomers) [93]. The resulting solid acid catalyst exhibited high efficiency in the synthesis of soluble cellulose derivatives. This was attributed to the enriching effect of its hollow structure, with a maximum separation yield of 73% for acetylated cellulose.

In 2021, Zhong’s team reported a study on hollow porous organic nanocatalysts (HPOF) with a multi-level porous shell structure, showcasing their efficient conversion of carbon dioxide [94]. The hollow interior of HPOF accelerated mass transfer of reactants to catalytic sites and facilitated the egress of products from the catalytic framework. Additionally, the layered porous thin shell exposed more active catalytic material, enhancing interactions with substrate molecules. The relationship between catalytic activity and the hollow structure was investigated through CO_2_ cycloaddition reactions, conducted without solvents, co-catalysts, or additives. As the thickness of the shell decreased, catalytic activity progressively increased. Notably, throughout the catalytic process, the selectivity of the desired product remained consistently above 99%.

In 2021, Gao et al. developed a cellular-structured solid acid catalyst known as SAPIL [95]. Depending on the composition of the reaction mixture, the concentration of ethyl acetate within SAPIL-1 was found to be 7.5 to 23.3 times higher than its external concentration. This observation indicates that SAPIL exhibits a significant enrichment capability towards ethyl acetate. The exceptional catalytic performance of SAPIL can be attributed to its 3D cellular structure in water and its high enrichment capacity for ethyl acetate. This paves the way for novel approaches in designing efficient heterogeneous acid catalysts.

In 2017, Huang et al. prepared a catalyst (Fe(TPP)Cl-MONN) with hierarchical porous structure, high specific surface area, and good stability through the combination of core-shell brush-like copolymers and Fe(TPP)Cl. The catalyst exhibited high catalytic activity and excellent reusability in the aldehyde olefination reaction and carbene insertion of N-H bond with diazoester. The hierarchical porous structure facilitated the accessibility of active sites and the diffusion of molecules, thus enhancing catalytic efficiency [96].

In 2020, Lin et al. synthesized an efficient heterogeneous catalyst using amine-functionalized mesoporous networked nanotube hyper-crosslinked bottlebrush copolymers (amine-MNNs) and Na_2_WO_4_ as raw materials [97]. The cavity-like structure traversed almost every nanotube in the network. They employed the oxidation of benzyl thioether as an activity test. The results demonstrated that Na_2_WO_4_·2H_2_O exhibited minimal catalytic activity for the reaction if not immobilized within the micropores of the networked nanotubes. This highlights the crucial impact of the nanotube structure on catalytic activity.

Huang’s team reported a method for preparing ultrafine Au NPs anchored on hollow porous organic nanospheres (Au@HPONs). The Au@HPONs nanocomposite exhibited excellent catalytic performance and cycling ability in the oxidation of benzyl alcohol and the reduction of 4-nitrophenol [98]. For comparison, the non-hollow structure Au@MOP catalyst showed lower conversion rates in the initial half-hour but reached a conversion rate as high as 99% after 2 h, demonstrating the importance of the hollow structure in accelerating the reaction rate (Figure 10). The cavity can facilitate the diffusion/transfer of substrates and products. The micropores of porous materials can be adjusted by swelling in different solvents, thereby affecting the mass transfer process. However, in pure microporous organic polymers, the matrix transition may become very slow with an increase in diffusion pathway. Conversely, this effect may not be significant in hollow structures with shorter diffusion pathways.

In 2019, Kim et al. employed Lewis acid–base interactions to guide the assembly of well-defined hierarchical nanostructured porous polymers [99]. Leveraging a rigid hollow polymer framework and inherent hydroxyl functionalities, the hyper-crosslinked hollow nanospheres were readily transformed into acid-functionalized catalysts. These were employed for efficient biodiesel production. Additionally, high-quality porous carbon nanomaterials, such as carbon nanotubes, hollow carbon nanospheres, and carbon nanosheets, were also produced through direct thermal decomposition of respective polymer precursors. The catalytic performance of solid acid catalysts was evaluated in the esterification reaction of fatty acids and methanol. The researchers found that the hollow spherical porous structure played a significant role in accelerating the reaction. Catalysts with such a hollow structure achieved high conversion rates for most substrates in just around 6 h at room temperature. The hollow spherical nanoporous polymers not only expanded their micropores through solvent swelling but also accelerated reactions by shortening the diffusion path of reactant molecules through their hollow structure.

In 2019, a team led by Li and Liu synthesized a two-dimensional nano-flower structured PS-b-PAA/Cu composite catalyst [100]. This composite membrane exhibited high catalytic activity for the reduction of p-nitroaniline in aqueous solution. The composite membrane also demonstrated good stability and high recyclability, with no changes in rate constant or conversion efficiency over 15 consecutive cycles. These characteristics were attributed to the unique structure of the composite nano-flowers. The hydrophilic outer layer of PAA facilitated the diffusion of reactants into the catalytic copper nanoparticles embedded within the PAA layer. The high stability could be attributed to the formation of thicker inner PS layers through the arrangement of petal-like nanosheets. The intertwined long PS chains within these layers enhanced the stability of the nano-flowers through strong hydrophobic interactions and intermolecular forces.

In 2020, Zhou et al. reported a novel catalyst with a hollow vesicle structure, where they found that the vesicle structure directly influenced the activity and performance of the catalyst [101]. Kinetic studies indicated that the unique vesicle structure allowed for substrate enrichment within a confined range, leading to an enhanced reaction rate. It was demonstrated that both the framework and morphology played a crucial role in the outstanding performance of the catalyst, particularly in the Suzuki–Miyaura cross-coupling reaction involving aryl chlorides.

In summary, the distribution of micropores and mesopores, along with the macroscopic morphology of materials, collectively determine their catalytic performance. These characteristics not only affect the mass transfer efficiency and selectivity of substrates but also play a crucial role in enhancing the contact with reactants and maximizing the specific surface area, thereby improving catalytic capabilities (Figure 11).

## 5. Conclusions and Perspective

In this review, the structure-oriented design of hyper-crosslinked polymer nanomaterials was discussed, summarizing the aspects of pore size distribution and macroscopic morphology. The design of micropores and mesopores can be achieved through the selection of suitable monomers, modification of crosslinking conditions, and the use of templates. Macroscopic morphology can be designed through templating methods, self-assembly methods, solvent weaving methods, and others. The microstructure affects the catalytic performance by influencing the mass transfer efficiency and the selectivity of reactants, while the macroscopic morphology affects the catalytic ability by influencing the accessibility with active sites and enrichment efficiency of reactants in the nanomaterials. By employing rational strategies, both macroscopic morphology and microstructure can be simultaneously designed to achieve better catalytic efficiency. Through these means, in future research, we can utilize the characteristics of hyper-crosslinked porous nanomaterials to design catalysts that are more favorable for catalytic reactions, thus obtaining optimal performance.

## Figures and Tables

**Figure 1 nanomaterials-13-02514-f001:**
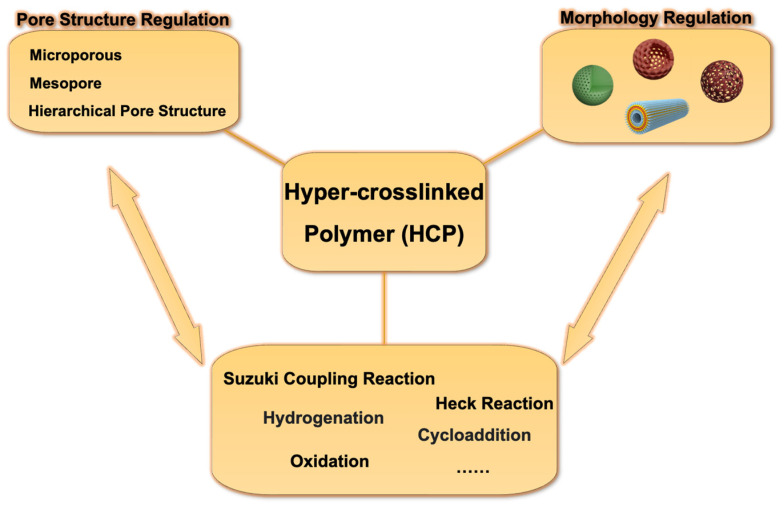
Pore size and morphology modulation of hyper-crosslinked polymer nanomaterials for catalytic applications.

**Figure 2 nanomaterials-13-02514-f002:**
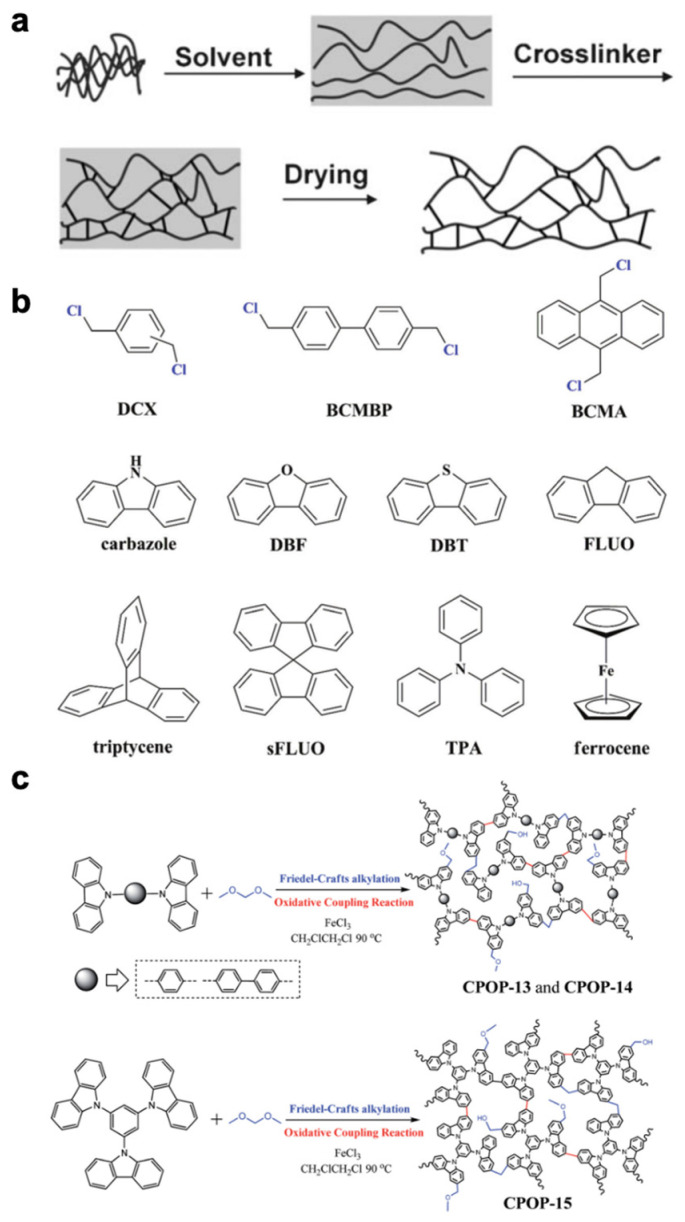
HCP synthesis methods: (**a**) post-crosslinking. Reprinted with permission from Ref. [42], Copyright 2007, Royal Society of Chemistry. (**b**) Direct crosslinking. Reprinted with permission from Ref. [45], Copyright 2017, Royal Society of Chemistry. (**c**) Crosslinking with the addition of a crosslinker. Reprinted with permission from Ref. [44], Copyright 2014, Royal Society of Chemistry.

**Figure 3 nanomaterials-13-02514-f003:**
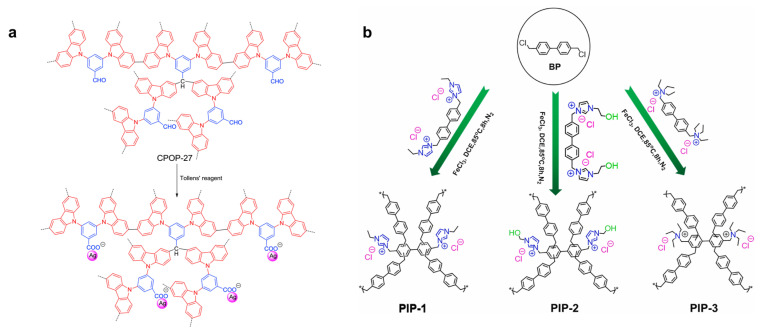
(**a**) Preparation of AgNPs/CPOP-27 composite. Reprinted with permission from Ref. [50], Copyright 2018, Elsevier. (**b**) Synthetic routes of PIPs. Reprinted with permission from Ref. [51], Copyright 2022, Elsevier.

**Figure 9 nanomaterials-13-02514-f009:**
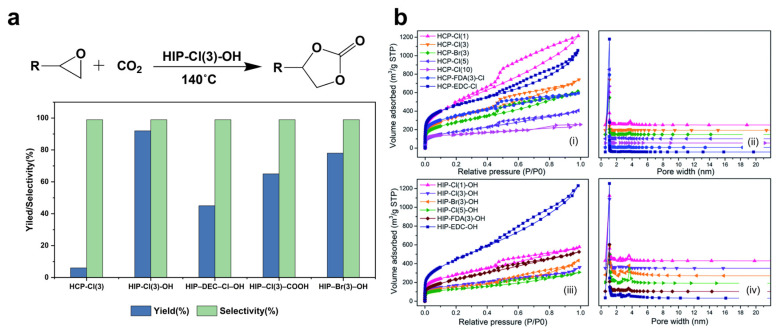
(**a**) Cycloaddition of CO_2_ with styrene oxide. (**b**) (**i**,**iii**) N_2_ adsorption–desorption isotherms and (**ii**,**iv**) pore size distribution of the polymers. Reprinted with permission from Ref. [92], Copyright 2022, Royal Society of Chemistry.

**Figure 10 nanomaterials-13-02514-f010:**
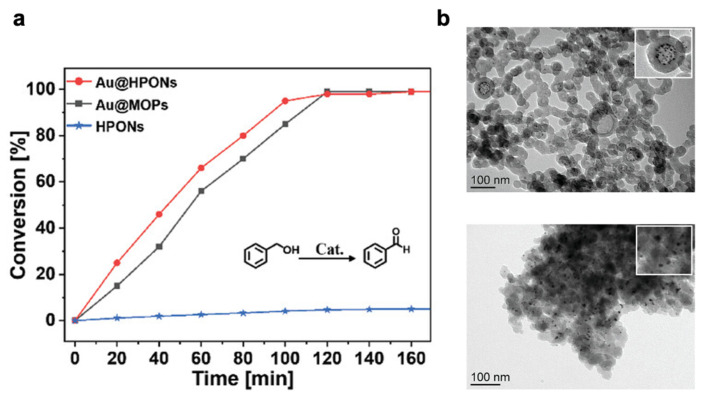
(**a**) The oxidations kinetic profiles of benzyl alcohol for Au@HPONs, Au@MOPs, and HPONs. (**b**) TEM images of Au@HPONs and Au@MOPs. Reprinted with permission from Ref. [98], Copyright 2023, John Wiley and Sons.

**Figure 11 nanomaterials-13-02514-f011:**
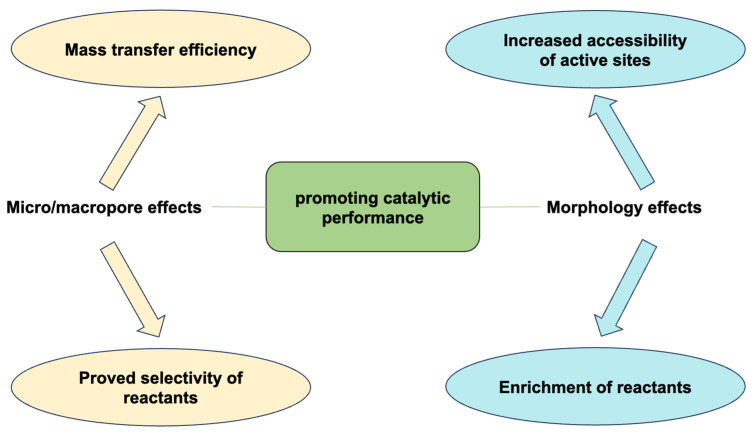
Influence of micro/mesopores and morphology on catalytic reactions.

## Data Availability

Not applicable.

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
