# Peer review of "Hyper-Crosslinked Porous Organic Nanomaterials: Structure-Oriented Design and Catalytic Applications"

_nanomaterials, 2023, doi:10.3390/nano13182514_

Round 1

Reviewer 1 Report

This is an interesting review concerning the structure-oriented design of hyper-crosslinked polymers. The synthesis of porous hyper-cross-linked polymers (HCPs) with increased specific surface area and porosity is an important topic with a large bibliography available including many studies focused on catalysis or adsorption and CO2 capture and environmental remediation… The morphology of nanomaterials can be essentially designed through templating methods while the tuning of the pore characteristics of HCPs can be achieved through the selection of suitable monomers, modification of crosslinking conditions, and the use of templates. The cited bibliography is nearly suitable to describe how the nanoparticle morphology and their porosity can lead to a, better catalytic efficiency. However, few very recent publications require to be cited as they are in the topic of the review notably the rational design of hyper-crosslinked polymers for biomedical applications (DOI: 10.1002/pol.20230270) or Hyper-crosslinked cyclodextrin porous polymer: an efficient CO2 capturing material with tunable porosity (https://doi.org/10.1039/C6RA18307G) or even Rational Fabrication of Polyethylenimine-Linked Microbeads for Selective CO2 Capture (https://doi.org/10.1021/acs.iecr.7b04212).

In conclusion, the present manuscript deserves to be published as it is of the main interest to the wider porous materials community. It provides interesting insights for nanomaterial structuration connected to required properties for the different applications of this type of nanoporous materials.

Line 36 Page 1 : bonding nature should be more appropriate than “bonding method” in the sentence [Currently, porous organic nanomaterials …. depending on their composition and bonding method.].

Reviewer 2 Report

The review is devoted to hyper-crosslinked porous polymers and authors tried to elucidate the relations between pore size and structure and catalytic performance of  catalysts based on them.

In the introduction the authors declared that in this review  “recent approaches to pore size modulation and morphology tailoring of HCPs and their applications in catalysis , with a focus on the effect of pore size modulation and morphology tailoring on catalytic application” are presented. Unfortunately, in the manuscript I did not find a comprehensive analysis of the studies on this issue. The approach to review used by authors is very similar to that used by Tan B. with co-authors in their reviews (“Recent Development of Hypercrosslinked Microporous Organic Polymers”, 2013; 10.1002/marc.201200788  and  “Hypercrosslinked porous polymer materials: design, synthesis, and applications”, 2017; 10.1039/C6CS00851H).  The advantage of the current review is that the manuscript contains the listing of recently published papers.

The illustrations for the review are overloaded.  See for example Figure 7. Often the pictures do not match a text. For example, Figure 6a and text in lines 216-223. Figure 8 should be modified as it contains different tables with catalysis results for different process and for unlike approaches to catalysts.

In my opinion the review needs a significant revision.

Moderate editing needs

Round 2

Reviewer 2 Report

The authors imporved the manuscript significantly, and I believe that now it can be pubslihed in Nanomaterials.